# Therapeutic Ultrasound Halts Progression of Chronic Kidney Disease In Vivo via the Regulation of Markers Associated with Renal Epithelial–Mesenchymal Transition and Senescence

**DOI:** 10.3390/ijms232113387

**Published:** 2022-11-02

**Authors:** Chen-Yu Lin, Ching-Chia Wang, Jui-Zhi Loh, Tsai-Chen Chiang, Te-I Weng, Ding-Cheng Chan, Kuan-Yu Hung, Chih-Kang Chiang, Shing-Hwa Liu

**Affiliations:** 1Institute of Toxicology, College of Medicine, National Taiwan University, Taipei 100, Taiwan; 2Department of Pediatrics, National Taiwan University Hospital, Taipei 100, Taiwan; 3Department of Forensic Medicine, College of Medicine, National Taiwan University, Taipei 100, Taiwan; 4Department of Geriatrics and Gerontology, College of Medicine, National Taiwan University, Taipei 100, Taiwan; 5Department of Internal Medicine, College of Medicine, National Taiwan University, Taipei 100, Taiwan; 6Departments of Integrated Diagnostics & Therapeutics and Internal Medicine, College of Medicine and Hospital, National Taiwan University, Taipei 100, Taiwan; 7Department of Medical Research, China Medical University Hospital, China Medical University, Taichung 404, Taiwan

**Keywords:** therapeutic ultrasound, chronic kidney disease, renal injury, senescence

## Abstract

Low-intensity pulsed ultrasound (LIPUS), a therapeutic type of ultrasound, is known to enhance bone fracture repair processes and help some tissues to heal. Here, we investigated the therapeutic potential of LIPUS for the treatment of chronic kidney disease (CKD) in two CKD mouse models. CKD mice were induced using both unilateral renal ischemia/reperfusion injury (IRI) with nephrectomy and adenine administration. The left kidneys of the CKD mice were treated using LIPUS with the parameters of 3 MHz, 100 mW/cm^2^, and 20 min/day, based on the preliminary experiments. The mice were euthanized 14 days after IRI or 28 days after the end of adenine administration. LIPUS treatment effectively alleviated the decreases in the body weight and albumin/globulin ratio and the increases in the serum renal functional markers, fibroblast growth factor-23, renal pathological changes, and renal fibrosis in the CKD mice. The parameters for epithelial–mesenchymal transition (EMT), senescence-related signal induction, and the inhibition of α-Klotho and endogenous antioxidant enzyme protein expression in the kidneys of the CKD mice were also significantly alleviated by LIPUS. These results suggest that LIPUS treatment reduces CKD progression through the inhibition of EMT and senescence-related signals. The application of LIPUS may be an alternative non-invasive therapeutic intervention for CKD therapy.

## 1. Introduction

Chronic kidney disease (CKD) is known to be a worldwide public health problem. Clinical wasting is an important risk factor for mortality in uremic patients, with a prevalence of 30–60% [1,2]. The prevalence of kidney disease cachexia in stages 1–2 and stages 3–5 of CKD has been estimated to be less than 2% and 11–46%, respectively [2]. The current evidence regarding the effective prevention of CKD remains insufficient. Reliable methods or novel treatment strategies are urgently needed for CKD/end-stage renal disease (ESRD) patients.

Low-intensity pulsed ultrasound (LIPUS) is a non-invasive therapeutic intervention. In 1983, Duarte showed that LIPUS could stimulate bone growth in a rabbit model [3]. In 1994, Heckman et al. reported a clinical application of LIPUS that could accelerate tibial fracture repair processes [4]. The US FDA for the treatment of established non-unions in 2000 approved LIPUS. Ogata et al. verified the efficacy of LIPUS treatment for cardiac dysfunction in a pressure-overloaded mouse heart model [5]. LIPUS was also found to be efficient against chronic myocardial ischemia in a porcine model [6]. Tang et al. also demonstrated that LIPUS could improve muscle atrophy in a type-one diabetic rat model [7]. However, the preventive effects of LIPUS application on CKD remain to be clarified.

The clinical manifestations of ischemia/reperfusion injury (IRI) or reoxygenation injury include conditions, such as acute kidney injury (AKI), myocardial infarction, organ transplantation, and major surgery [8]. AKI recovery is often incomplete. With the gradual loss of renal function over time, AKI predisposes patients to CKD [9]. The prevalence of CKD has increased over recent years. Increasing amounts of evidence have suggested that pathological factors, such as hypoxia, inflammation, and renal cell injury, play important roles in the progression of CKD [10]. The proliferation of fibroblasts and the excessive deposition of extracellular matrices, including collagen, in renal tissues after IRI contribute to renal fibrosis and CKD progression [11]. Renal fibrosis is irreversible and can be life-threatening. Pulsed ultrasound was demonstrated to possess therapeutic potential for preventing AKI in IRI-induced AKI mouse models [12,13]. However, there is still a lack of suitable therapeutic interventions that can delay CKD progression.

In this study, we investigated the protective effects of LIPUS treatment for CKD in vivo. CKD was induced using two mouse models of IRI and adenine administration, which cause CKD through different pathogenic mechanisms. These models were used to verify the protective effects of LIPUS on CKD. In clinical applications, the intensity range of therapeutic ultrasound is generally set to 30–1000 mW/cm^2^ [14]. Our previous study showed that LIPUS at a 100 mW/cm^2^ intensity effectively improved H_2_O_2_- and hypoxia/reoxygenation-induced acute renal cell injury in vitro and physiopathological alterations induced by AKI in vivo, but not LIPUS at a 30 mW/cm^2^ [13]. Our preliminary experiments also tested the effects of LIPUS on CKD mice with various parameters based on previous studies [13,14]. Finally, we selected the parameters of 3 MHz, 100 mW/cm^2^, and 20 min/day for our formal experiments.

## 2. Results

### 2.1. LIPUS Alleviated Changes in Body Weight, Serum Biochemistry, and Renal Injury in IRI-CKD Mouse Models

Firstly, the unilateral IRI was performed via contralateral nephrectomy on CKD mouse models after 14 days with or without LIPUS treatment. As shown in Figure 1A, both body weight and relative kidney weight significantly decreased in the CKD mice, which could be partially reversed by LIPUS treatment. The serum albumin/globulin ratio also significantly decreased (Figure 1B), and serum creatinine, BUN, and cystatin C levels markedly increased (Figure 1C) in the CKD mice, which could be significantly reversed by LIPUS treatment. 

LIPUS treatment could also significantly reverse the histopathological changes from renal injury (Figure 2A) and renal collagen deposition stained with Masson’s trichrome (Figure 2B) in the kidneys of the CKD mice. The levels of serum FGF-23, a biomarker of phosphorus metabolism during CKD, markedly increased in the IRI-induced CKD mice, which could also be reversed by LIPUS treatment (Figure 2C). We further tested for changes in the levels of serum indoxyl sulfate. Serum indoxyl sulfate levels markedly and significantly increased in the IRI-CKD mice, which could be significantly reduced by LIPUS treatment (Figure 2D).

We further found that the fibrosis markers of α-SMA, E-cadherin, and vimentin protein expression were markedly enhanced in the kidneys of the IRI-induced CKD mice, which could be reversed by LIPUS treatment (Figure 3A). Moreover, apoptosis-related signaling molecules, such as CHOP, Bax, Bcl-2, and cleaved caspase 3, were also induced in the kidneys of the IRI-CKD mice. Still, LIPUS significantly inhibited the induction of these apoptosis-related signaling molecules (Figure 3B). We also tested the involvement of oxidative stress in CKD mice. The protein expression levels of SOD1 and catalase decreased in the kidneys of the IRI-induced CKD mice, and LIPUS treatment reversed the decrease in SOD1 and catalase protein expression (Figure 3C). Maladaptive repair processes after AKI have been suggested to contribute toward the development of kidney ageing and CKD [15]. We further found that the protein expression levels of senescence-related molecules p53, p16, p21, and Sirt1 were upregulated and that α-Klotho, a putative aging-suppressor protein, was downregulated in the kidneys of the IRI-induced CKD mice, which could be reversed by LIPUS treatment (Figure 3D).

We further examined the involvement of inflammatory cell infiltration and cytokine/growth factor expression in the kidneys of the IRI-CKD mice with and without LIPUS treatment. The protein expression levels of Ly6G (a neutrophil marker), CD68 (a macrophage marker), and inflammatory cytokine TNF-α were significantly enhanced in the kidneys (Figure 4A) of the IRI-CKD mice, but LIPUS could effectively and significantly inhibit these inflammatory markers (Figure 4A). Moreover, the levels of mRNA expression for IL-6 and TGF-β (*p* < 0.05) were also significantly increased in the kidneys of the IRI-CKD mice, but not those for IL-1β (*p* = 0.072). LIPUS could also effectively reverse the increased IL-6 and TGF-β mRNA expression (Figure 4B).

### 2.2. LIPUS Alleviated Changes in Body Weight, Serum Biochemistry, and Renal Injury in Adenine-Induced CKD Mouse Models

CKD was induced in another mouse model via adenine administration (50 mg/kg) for 28 days in the presence or absence of LIPUS treatment. Histological changes in the kidneys of adenine-treated mice are characterized by interstitial fibrosis [16]. We found that body weight significantly decreased at day 7 of adenine administration in the mice, which could be reversed by LIPUS treatment on day 21 and day 28 (Figure 5(Aa and b)). The relative weights of both the left and right kidneys increased in the adenine-CKD mice, which could be reversed by LIPUS treatment (Figure 5B). The levels of serum creatinine and BUN markedly increased in the adenine-CKD mice, which could be reversed by LIPUS treatment at day 21 (BUN: sham, 28.62 ± 3.92, adenine, 127.53 ± 24.45, adenine + LIPUS, 44.84 ± 5.96 mg/dL, n = 5, *p* < 0.05; creatinine: sham, 0.33 ± 0.02, adenine, 0.98 ± 0.09, adenine + LIPUS, 0.49 ± 0.06 mg/dL, n = 5, *p* < 0.05) and day 28 (Figure 5C). The serum albumin levels significantly decreased, serum globulin levels significantly increased, and serum albumin/globulin ratio significantly decreased in the adenine-CKD mice, all of which could be partially reversed by LIPUS treatment (Figure 5D). Our observations of gross appearance showed that both the left and right kidneys in the adenine group were coarse and pale, while both kidneys in adenine + LIPUS group appeared similar to those in the normal control group (Figure 6A). The renal histopathological score markedly increased in the adenine-CKD mice, which could be reversed by LIPUS treatment (Figure 6B). Moreover, the levels of serum indoxyl sulfate also markedly and significantly increased in the adenine-CKD mice, which could be significantly reduced by LIPUS treatment (Figure 6C). We further found that the protein expression levels of senescence-related molecules p53, p21, and p16 were upregulated in the left kidneys of the adenine-CKD mice, which could be reversed by LIPUS treatment (Figure 6D).

Renal collagen deposition markedly increased in the left kidneys of the adenine-CKD mice, which could be reversed by LIPUS treatment (Figure 7A). We further found that the protein expression of the fibrosis marker α-SMA and epithelial–mesenchymal transition (EMT) activation (i.e., decreased E-cadherin and increased vimentin protein expression) were markedly enhanced in both the left (Figure 7(Ba)) and right (Figure 7(Bb)) kidneys of the adenine-CKD mice, which could be reversed by the LIPUS treatment applied to the left kidneys. 

## 3. Discussion

In the present study, we investigated the protective potential of therapeutic ultrasound against CKD using the IRI-induced and adenine-induced CKD mouse models. In the IRI-CKD mouse model, ischemia was induced in the left kidneys and a nephrectomy was performed on the right kidneys 24 h before the mice were euthanized [17,18]. The study of Shu et al. showed that the serum creatinine levels, renal histological score, and protein expression levels of fibrosis markers significantly increased in the blood samples and the left kidney tissues, which were collected 2, 7, and 14 days after left kidney ischemia; the nephrectomy was performed on the right kidneys one day before euthanasia [18]. Our results showed that LIPUS treatment reduced CKD progression by inhibiting EMT and senescence.

Growth factors, such as VEGF and TGF-β, derived from mesenchymal stem cells have been demonstrated to recruit leukocytes and repair intrinsic cells via paracrine signals and extracellular vesicles, which may be involved in AKI repair or AKI–CKD transition [19]. TGF-β1 mRNA levels have been found to increase from 12–24 h to 14 days after IRI in rats [20]. TGF-β can trigger renal fibrosis, potentially through its action on macrophage chemoattraction; however, fibrosis may possibly be induced by macrophages via the production of cytokines other than TGF-β1 [21]. Guerrero and McCarty indicated that TGF-β possessed angiogenic and angiostatic properties under physiological and pathological conditions during vascular development, based on its expression content and tissue/organ context [22]. Kinashi et al. mentioned that TGF-β triggered peritoneal dialysis-associated peritoneal fibrosis and neo-angiogenesis via VEGF-A interaction and that a TGF-β/VEGF-C pathway could participate in the renal and peritoneal fibrosis-associated lymph-angiogenesis [23]. TGF-β-induced renal tubular cell apoptosis has been suggested to be due to cell cycle arrest rather than direct pro-apoptotic action [24]. Therefore, TGF-β signaling may participate in renal inflammation, apoptosis, and angiogenesis. The present study also found an increase in the renal TGF-β expression in the IRI-CKD mice, which could be reversed by LIPUS treatment. However, the detailed effects and mechanisms of LIPUS on renal angiogenesis, apoptosis, and inflammation in the IRI-CKD mouse model need to be clarified in the future.

In a cohort study of CKD patients, the levels of serum indoxyl sulfate were found to be related to vascular alterations and mortality [25]. A positive relationship between the levels of serum indoxyl sulfate and CKD progression in pediatric patients has also been reported [26]. Yang et al. found that the increased serum indoxyl sulfate-induced platelet hyperactivity led to thrombosis in CKD mice [27]. In the present study, we also observed an increase in the levels of serum indoxyl sulfate in the CKD mice, which could be effectively decreased by LIPUS treatment, implying the possible pathological role of indoxyl sulfate in CKD mice.

The prevalence of kidney cachexia increases when renal function declines [28]. A previous study found that stage 3–5 CKD patients with a low albumin/globulin ratio had the worst overall survival rates, while patients with a high albumin/globulin ratio had the best overall survival rates [29]. Circulating FGF-23 levels progressively elevate in patients with CKD. High FGF-23 levels have been suggested to be strongly correlated with serum creatinine levels and associated with kidney disease progression [30]. Elevated FGF-23 levels may induce inflammation, leading to protein-energy wasting/cachexia [30]. The present study found that LIPUS treatment significantly alleviated decreased serum albumin/globulin ratios and increased serum BUN, creatinine, cystatin C, FGF-23, and indoxyl sulfate levels and renal injury/fibrosis in the CKD mice. These findings suggested that LIPUS treatment could potentially prevent CKD-associated renal injury.

Our results also showed that catalase and SOD1 protein expression levels in the kidneys of the IRI-associated CKD models could be restored by LIPUS treatment. Renal ischemia/reperfusion can degrade cellular ATP into hypoxanthine that is further converted into xanthine by xanthine oxidase and produces a superoxide radical, which is converted into H_2_O_2_ by SOD and then the H_2_O_2_ can be converted into water and oxygen by catalase [31]. Moreover, Kobayashi et al. found that catalase deficiency deteriorated renal oxidant tissue injury and triggered EMT in remnant kidneys and renal fibrosis progression in a CKD mouse model [32]. These findings indicated that systems for cell antioxidant defense could be retrieved by LIPUS via protecting cells from further ROS damage in CKD mice.

The remarkable effects of LIPUS treatment on inhibiting increases in the renal p53, p16, and p21 protein expression and the serum levels of FGF-23 and restoring deficiencies in the renal Klotho protein after IRI-associated CKD were shown in this study. Senescence growth arrest is known to be regulated by two tumor-suppressing pathways: p53/p21 and p16 [33]. The role of p53 in renal IRI has been confirmed by observing the reduced short-term consequence of renal IRI and preventing the long-term consequence of interstitial fibrogenesis in p53 knockout mice [34]. 

Moreover, the Klotho gene, which encodes the α-Klotho protein, has been recognized as a putative aging-suppressor gene. Additionally, the α-Klotho levels decrease in patients with AKI or CKD and in animal models after IRI [35]. Klotho deficiency triggers a higher susceptibility toAKI, delays renal regeneration, and enhances renal fibrosis [36]. Klotho has been found to protect renal tubular cells from IRI by inhibiting oxidative stress [35]. Moreover, the membrane Klotho protein is known to act as a co-receptor for FGF-23, which participates in mineral homeostasis [36]. Elevated FGF-23 levels lower Klotho expression, so its suppressing effect on klotho production can induce vascular and metabolic disorders [30]. The results of the present study suggest that LIPUS treatment can prevent the induction of renal cell senescence and delay the development of fibrogenesis during CKD progression.

The role of SIRT1, a NAD+-dependent protein deacetylase, in renoprotection is controversial. SIRT1 protein expression markedly increases in the mouse kidneys with IRI-associated CKD. The renoprotection induced by SIRT1 is closely involved in the inhibition of the p53 signaling pathway in young mice with IRI-associated AKI [37]. However, the role of SIRT1 in renal injury and fibrogenesis is controversial. The inhibition of SIRT1/2 has been shown to suppress renal fibroblast activation and proliferation and attenuate renal fibrogenesis in obstructive nephropathy [38]. Muratsubaki et al. recently found that SIRT1 expression was markedly upregulated in the kidneys of non-diabetic rats with IRI-associated AKI but not in those of diabetic rats [39]. In the present study, the elevation of SIRT1 expression in IRI-associated CKD could have played a role in promoting renal fibrogenesis, which could have been reversed by LIPUS.

The present study demonstrated for the first time that LIPUS with a 3 MHz frequency and 0.1 W/cm^2^ intensity (20 min daily) effectively protected against CKD-related renal injury in experimental IRI- and adenine-induced CKD mouse models. LIPUS is known as a therapeutic ultrasound with a frequency range of 0.7–3 MHz that is delivered at a lower intensity (<3 W/cm^2^) than traditional ultrasound energy [14]. LIPUS with a frequency of 3 MHz and an intensity of 2.2 W/cm^2^ for 15 min/day for two days was shown to reduce sperm counts in rats, which indicated that LIPUS could be a candidate for male contraceptive therapy [40]. LIPUS with the parameters of 3 MHz, 0.1 W/cm^2^, and 20 min/day for seven days was shown to prevent inflammatory responses in an IRI-induced AKI mouse model [13]. The diameter of the Sonicator^®^-740 ultrasound probe used in this study was about 1.13 cm, which applied an effective radiating area of 1 cm^2^ to the exposed site. When the ultrasound probe was applied to the kidney of a male adult mouse whose kidney was about 1.07 cm in size [41], the kidney could be coupled with a transducer. According to the instrument manufacturer’s instruction manual, this pencil-shaped 1 cm^2^ ultrasound probe can clinically deliver ultrasound to hard-to-reach body parts and smaller treatment areas.

Our results for the adenine-CKD mouse model showed that LIPUS stimulation on the left kidney could effectively improve lesions on both the left and right kidneys. These findings suggest that there can be endogenous protective factors transmitted from the left kidney to the right kidney. Gigliotti et al. have tested the protective effects of therapeutic ultrasound on renal injury in a bilateral IRI-induced AKI mouse model. They found that it had a markedly renal protective effect through a splenic anti-inflammatory pathway when ultrasound only stimulated the left kidney [12,42]. The protective mechanisms of LIPUS treatment, including the splenic anti-inflammatory pathway and others, on the right kidneys of adenine-CKD mice when only the left kidneys are stimulated need further investigation.

In the present study, the mice were pretreated with LIPUS before CKD induction. In our preliminary tests, when LIPUS was administered after CKD induction in mice, the results showed that the therapeutic efficacy was not as successful as when LIPUS was administered beforehand. This could be a LIPUS parameter-setting problem, and we have not yet found an appropriate parameter for post-treatment with LIPUS. We hope to continue to explore this issue in the future. 

There are several study limitations: (1) It is very likely that the LIPUS therapy targeting the left kidney in both CKD models is exclusive or also effective via the splenic anti-inflammatory pathway, as reported by Okusa et al. [12,42]; (2) As the LIPUS effect is only documented at 14 and 28 days in the IRI and adenine models, respectively, no long-term data are obtained to determine whether the observed beneficial effects are permeant or only transient; (3) The status of the renal microvasculature/vasoprotection is also important for LIPUS treatment. These issues are left for future investigation.

## 4. Materials and Methods

### 4.1. Animals and Experimental Protocol

Adult (six-week-old) male C57BL/6J mice were used, provided by the Laboratory Animal Centre of the National Taiwan University College of Medicine, where the animal experiments took place. The mice were housed in cages with a temperature of 22 ± 2 °C and 12 h light/dark cycles. The mice had access to standard rodent feed and water. The animal experiments were approved by the Institutional Animal Care and Use Committee of the National Taiwan University College of Medicine and were performed in accordance with the ethical standards laid down in the 1964 Declaration of Helsinki and its later amendments. The animal experiments followed Taiwan regulations and NIH guidelines to ensure animal welfare and reduce animal suffering.

#### 4.1.1. IRI-Induced CKD Model

The mice were randomly divided into control, IRI, and IRI + LIPUS groups (eight mice per group). Before the experiments, the mice were anesthetized with ketamine (100 mg/kg, i.p.) and xylazine (10 mg/kg, i.p.). To induce IRI, a non-traumatic artery clamp was used to clamp the left renal artery for 30 min, followed by reperfusion. The mice were placed on a heat plate, and the rectal temperature was maintained at 37 ± 0.5 °C during the experiments and recovery from anesthesia for both the IRI and control procedures. The right kidney was nephrectomized 24 h before euthanasia [17,18]. In the IRI + LIPUS group, pre-treatment with LIPUS was performed five days before the IRI procedure. LIPUS treatment was also continuously applied after IRI until the day of euthanasia. The mice were euthanized 14 days after IRI. The mice in the control group underwent all surgery steps except for the artery clamping.

#### 4.1.2. Adenine-Induced CKD Model 

CKD was induced in the adenine mouse model as previously described [16]. The mice were randomly divided into control, adenine, and adenine + LIPUS groups (five mice per group). In the control group, 0.5% CMC was administered by oral gavage. In the adenine group, adenine (50 mg/kg) in 0.5% CMC was administered by oral gavage. In the adenine + LIPUS group, mice were pre-treated with LIPUS for five days before adenine administration, and LIPUS treatment was continuously applied after adenine administration until the day of euthanasia. The kidneys were harvested and weighed. The tissues were fixed in 10% formalin at room temperature for histological analysis. Portions of these samples were frozen at –80 °C for Western blot analysis.

### 4.2. LIPUS Treatment 

A clinically available and portable therapeutic ultrasound device (Sonicator^®^-740 from Mettler Electronics, Anaheim, CA, USA) with a 3 MHz single-element focused transducer (1 cm^2^ diameter) was used to generate the LIPUS. During the daily LIPUS application, anesthesia was administered to the mice (isoflurane (Tokyo Chemical Industries, Ltd., Tokyo, Japan) mixed with 3% oxygen). The LIPUS transducer was placed on the left lateral regions of the abdomens of the mice under anesthesia, and the signal was transmitted through coupling gel. The left kidneys of the IRI- or adenine-induced mice were exposed to LIPUS for 20 min/day with the aforementioned parameters, including a spatial peak temporal average intensity of 100 mW/cm^2^, a pulse repetition rate of 100 Hz with a pulse width of 5 ms and 5 ms between pulses (pulse space = 10 ms), and a 50% duty cycle with a 1:1 ratio of time to off-time. These parameters were chosen to obtain minimal thermal effects [13,14]. The LIPUS parameters used were selected based on the existing literature [13,14] and our preliminary study.

### 4.3. Serum Biochemistry Analysis

A commercially available clinical chemistry analyzer (Roche, Rotkreuz, Switzerland) was used to measure the serum albumin, globulin, creatinine, and blood urea nitrogen (BUN) levels. A mouse ELISA kit (Immunology Consultants Laboratory, Portland, OR, USA) was used to detect the levels of cystatin C. Serum fibroblast growth factor (FGF)-23 was determined using a mouse FGF23 ELISA kit (Abcam, Cambridge, MA, USA).

### 4.4. Detection of Serum Indoxyl Sulfate

Blood samples were collected from a puncture in the submandibular vein and allowed to clot at room temperature for 30 min. The samples were centrifuged for 15 min at 3000 rpm (1500× *g*) and 4 °C for serum collection. The levels of serum indoxyl sulfate were detected by ultra-high performance liquid chromatography (UHPLC; Agilent 1290 infinity II, Santa Clara, CA, USA) and tandem mass spectrometry (LC-MS/MS; Sciex QTRAP 6500, Framingham, MA, USA), assisted by Dr. Te-I Weng from the Department of Forensic Medicine, College of Medicine, National Taiwan University. Briefly, 50 µL of serum was mixed with distilled deionized water and internal standard (in acetonitrile) and was then incubated at –20 °C for 30 min for protein precipitation. The samples were centrifuged for 10 min, and the supernatant was collected and dried using nitrogen blowdown. Then, the samples were reconstituted using 10% methanol and analyzed. 

### 4.5. Histology Analysis 

The kidney tissues were embedded in paraffin. The 4 μm thick tissue sections were stained with Periodic acid–Schiff (PAS) and Masson’s trichrome to analyze histological changes and fibrosis, respectively. Then, 15 randomly selected fields per section were analyzed. Scores from 0–4 were used to analyze tubular injuries, such as renal tubule dilation, tubular epithelial injury, and cast formation, which were then graded by a veterinary pathologist for a blind analysis: a score of 0 represented no change; a score of 1 represented a change affecting <25% of the field; a score of 2 represented a change affecting 25–50% of the field; a score of 3 represented a change affecting 50–75% of the field; a score of 4 represented a change affecting >75% of the field. 

### 4.6. Immunoblotting 

The protein expression levels were determined by Western blotting. The samples were re-labeled for a blind analysis before the Western blot processing. Equal amounts of protein extracts were separated using sodium dodecyl sulfate (SDS)-PAGE and then electrophoretically transferred onto polyvinylidene difluoride (PVDF) membranes. The membranes were incubated with the primary antibodies for E-cadherin (#866702), vimentin (#677802), α-SMA (#904601) (Biolegend, San Diego, CA, USA), p53 (#sc-126), p16 (#sc-377412), p21 (#sc-6246), β-actin (#sc47778), GAPDH (#sc25778) (Santa Cruz Dallas, TX, USA), catalase (#ab16731), superoxide dismutase 1 (SOD1; #ab13498), Klotho (#ab203576), CD68 (#ab125212), TNF-α (#ab183218) (Abcam, Cambridge, MA, USA), Ly6G (#14-5931-82) (eBioscience, San Diego, CA, USA), Bax (#14796), Bcl-2 (#3498), cleaved Caspase 3 (#9664), and CHOP (#2895) (Cell Signaling Technology, Danvers, MA, USA), followed by incubation with horseradish peroxidase-conjugated secondary antibodies (Bio-Rad, Hercules, CA, USA). The blot signals were measured using enhanced chemiluminescence substrates (Bio-Rad) and developed onto a Fuji Blue X-ray Film. ImageJ software was used to quantify the protein expression bands.

### 4.7. Real-Time Quantitative PCR (qPCR)

The kidney gene expression was analyzed using qPCR, as previously described in [13]. Briefly, total RNA (5 µg) was reversely transcribed into cDNA using a commercial kit (Thermo Fisher Scientific, Waltham, MA, USA). Then, 100 ng of cDNA products was applied as a template for amplification using the SYBR Green PCR master mix (Thermo Fisher Scientific) and gene-specific primers (IL1-β forward: 5′-AGTTGACGGACCCCAAAAG-3′; IL1-β reverse: 5′-AGCTGGATGCTCTCATCAGG; IL-6 forward: 5′-GCTACCAAACTGGATATAATCAGGA-3′; IL-6 reverse: 5′-CCAGGTAGCTATGGTACTCCAGAA-3′; TGF-β forward: 5′-TGGAGCAACATGTGGAACTC-3′; TGF-β reverse: 5′-GTCAGCAGCCGGTTACCA-3′; Gapdh forward: 5′-AAGAGGGATGCTGCCCTTAC-3′; Gapdh reverse: CCATTTTGTCTACGGGACGA-3′). The gene expression was detected using an Applied Biosystems StepOne™ Real-Time PCR System (Thermo Fisher Scientific). The tested mRNA expression levels were then normalized using Gapdh. 

### 4.8. Statistics

The results are expressed as the mean ± SD. A one-way analysis of variance (ANOVA) was used to assess significant differences, followed by an unpaired two-tailed Student’s *t*-test, and *p*-values < 0.05 were considered statistically significant. The GraphPad Prism 6 software was used for the data graphing and statistical analysis.

## 5. Conclusions

Our findings demonstrated that LIPUS treatment could protect against kidney disease cachexia by ameliorating CKD progression via the inhibition of renal fibrosis, the restoration of antioxidant enzymes, the attenuation of renal senescence/aging, and decreases in uremic toxin indoxyl sulfate levels (Figure 7C). Based on these findings, LIPUS may be considered as an alternative non-invasive therapeutic intervention to treat kidney disease or serve as an auxiliary tool for the management of CKD.

## Figures and Tables

**Figure 1 ijms-23-13387-f001:**
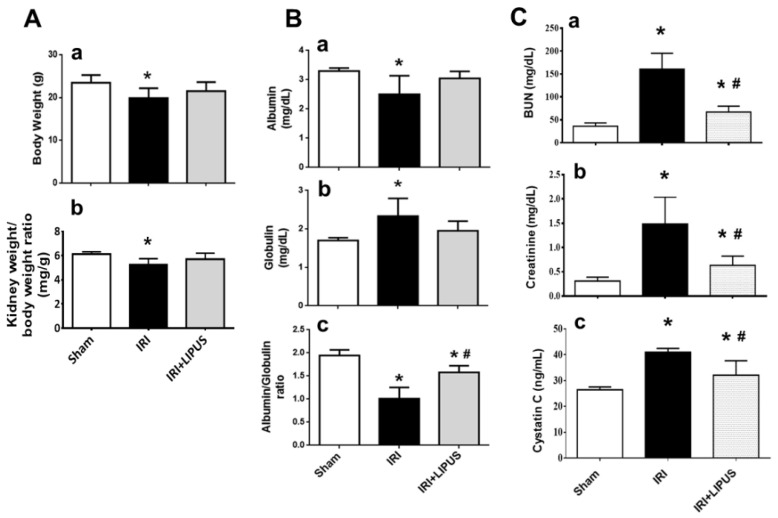
The effects of LIPUS on body weight, kidney weight, and serum biochemistry following unilateral IRI via contralateral nephrectomy on CKD mouse models (the mice were euthanized 14 days after IRI): (**A**) body weight (**a**) and kidney weight (**b**); (**B**) serum albumin levels (**a**), globulin levels (**b**), and albumin/globulin ratio (**c**); (**C**) serum blood urea nitrogen levels (BUN, (**a**)), creatinine levels (**b**), and cystatin C levels (**c**). Data are presented as mean ± SD (n = 8): * *p* < 0.05 versus the control group; # *p* < 0.05 versus the CKD-alone group.

**Figure 2 ijms-23-13387-f002:**
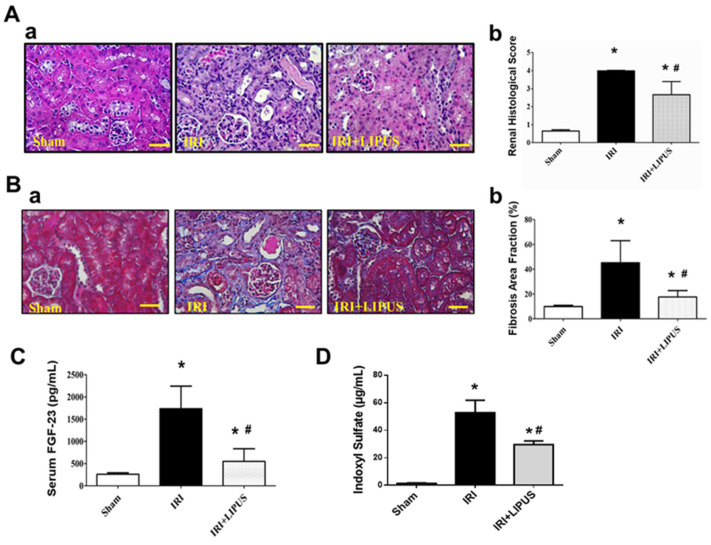
The effects of LIPUS on renal histology, renal fibrosis, and serum FGF-23 and indoxyl sulfate levels in IRI-CKD mice (the mice were euthanized 14 days after IRI): (**A**) (**a**) pathological changes in renal tissue sections stained with Periodic acid–Schiff (PAS), which were analyzed using a light microscope, and (**b**) histological score (scale bar: 50 μm); (**B**) (**a**) renal collagen deposition stained by Masson’s trichrome and (**b**) quantifications (scale bar: 50 μm); (**C**) serum FGF-23 levels; (**D**) serum indoxyl sulfate levels. Data are presented as mean ± SD (n = 8): * *p* < 0.05 versus the control group; # *p* < 0.05 versus the CKD-alone group.

**Figure 3 ijms-23-13387-f003:**
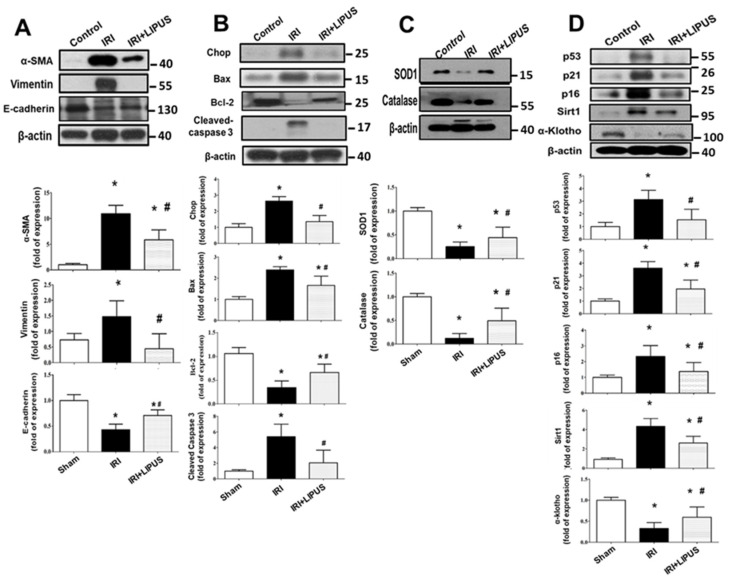
The effects of LIPUS on fibrosis-related signaling molecules, endogenous antioxidant enzymes, and senescence-related molecular signals in the kidneys of IRI-CKD mice (the mice were euthanized 14 days after IRI): the protein expression levels for fibrosis-related signaling molecules (α-SMA, E-cadherin, and vimentin) (**A**); the protein expression levels for apoptosis-related signaling molecules (CHOP, Bax, Bcl-2, and cleaved caspase 3) (**B**); the protein expression levels for antioxidant enzymes SOD1 and catalase (**C**); the protein expression levels for senescence-related molecules (p53, p21, p16, Sirt1, and α-Klotho) (**D**). These levels in the kidneys were determined by Western blot. Quantifications for the protein expression are also shown. Data are presented as mean ± SD (n = 4–6): * *p* < 0.05 versus the control group; # *p* < 0.05 versus the CKD-alone group.

**Figure 4 ijms-23-13387-f004:**
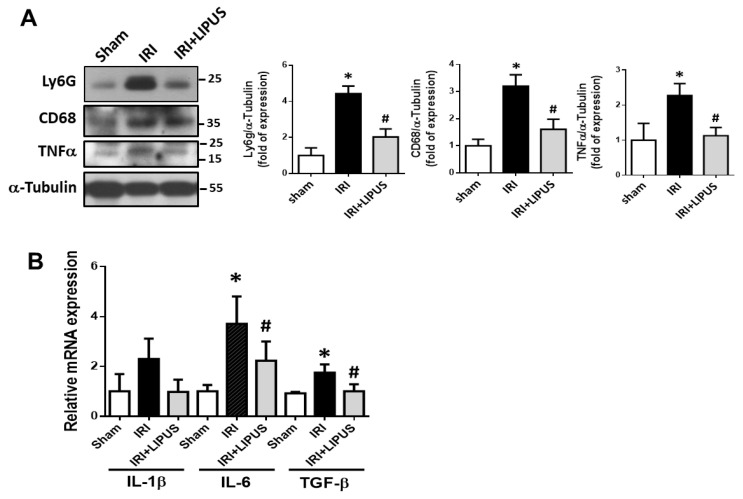
The effects of LIPUS on inflammatory cell infiltration in the kidneys of IRI-CKD mice: Ly6G, CD68, and TNF-α protein expression levels in the kidneys, determined by Western blot (**A**); IL-1β, IL-6, and TGF-β mRNA expression levels in the kidneys, determined by real-time PCR (**B**). Data are presented as mean ± SD (n = 4–6): * *p* < 0.05 versus the control group; # *p* < 0.05 versus the IRI-CKD group.

**Figure 5 ijms-23-13387-f005:**
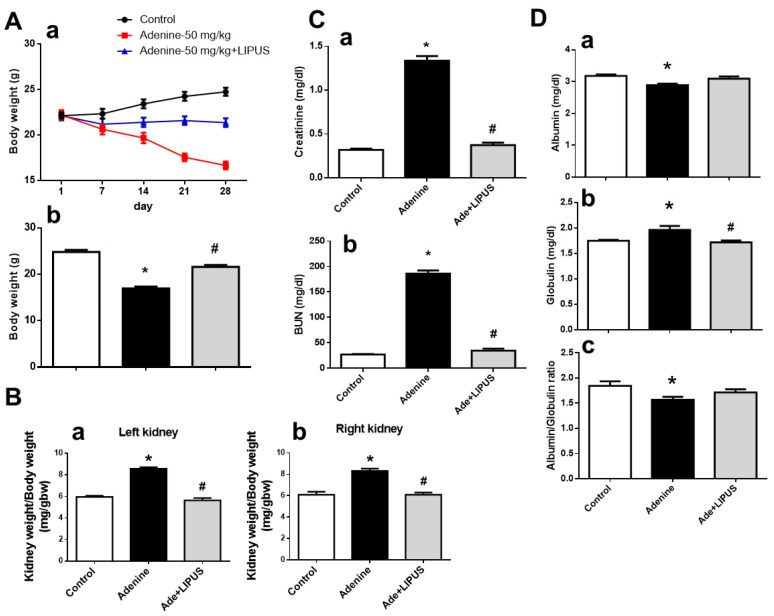
The effects of LIPUS on changes in body weight, relative kidney weight, and serum biochemistry in CKD mouse models following adenine administration (the mice were euthanized 28 days after adenine administration): (**A**) changes in body weight over time (**a**) and final body weight (**b**); (**B**) the relative weights of the left kidneys (**a**) and right kidneys (**b**); (**C**) serum blood urea nitrogen levels (BUN, (**a**)) and creatinine levels (**b**); (**D**) serum albumin levels (**a**), globulin levels (**b**), and albumin/globulin ratio (**c**). Data are presented as mean ± SD (n = 9): * *p* < 0.05 versus the control group; # *p* < 0.05 versus the CKD-alone group.

**Figure 6 ijms-23-13387-f006:**
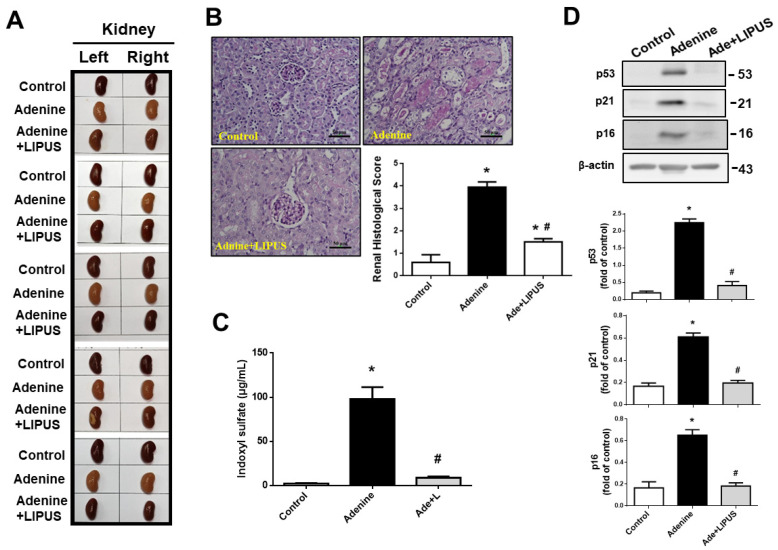
The effects of LIPUS on renal histology and serum indoxyl sulfate levels in CKD mouse models following adenine administration (the mice were euthanized 28 days after adenine administration): (**A**) the gross appearance of both the left and right kidneys; (**B**) pathological changes in renal tissue sections from the left kidneys stained with Periodic acid–Schiff (PAS), which were analyzed using a light microscope, and histological score (scale bar: 50 μm); (**C**) serum indoxyl sulfate levels; (**D**) senescence-related molecules (p53, p21, and p16) in the left kidneys, determined by Western blot. Quantifications for the protein expression are also shown. Data are presented as mean ± SD (n = 5–9): * *p* < 0.05 versus the control group; # *p* < 0.05 versus the CKD-alone group.

**Figure 7 ijms-23-13387-f007:**
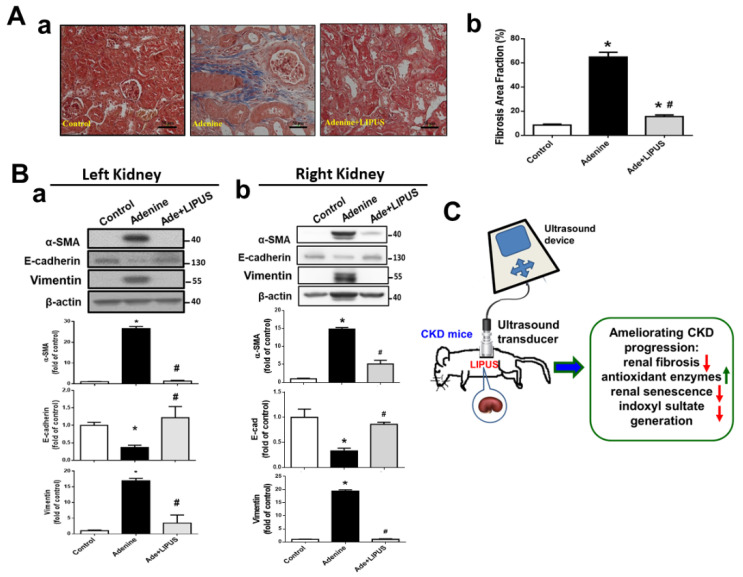
The effects of LIPUS on renal fibrosis and fibrosis-related signaling molecules in CKD mouse models following adenine administration: (**A**) (**a**) renal fibrosis stained by Masson’s trichrome in the left kidneys and (**b**) quantifications (scale bar: 50 μm; data are presented as mean ± SD (n = 10); * *p* < 0.05 versus the control group; # *p* < 0.05 versus the CKD-alone group); (**B**) protein expression levels for α-SMA, E-cadherin, and vimentin in both left (**a**) and right (**b**) kidneys of the adenine-induced CKD mice, determined by Western blot, and quantifications for the protein expression (data are presented as mean ± SD (n = 4–6); * *p* < 0.05 versus the control group; # *p* < 0.05 versus the CKD-alone group); (**C**) a schematic summary of our main findings regarding the effects of LIPUS on CKD-associated renal injury. Green upward arrows represent up-regulation. Red downward arrows represent down-regulation.

## Data Availability

The data presented in this study are available from the corresponding author upon reasonable request.

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
