# Peer review of "Therapeutic Ultrasound Halts Progression of Chronic Kidney Disease In Vivo via the Regulation of Markers Associated with Renal Epithelial–Mesenchymal Transition and Senescence"

_ijms, 2022, doi:10.3390/ijms232113387_

Round 1
Reviewer 1 Report
Dr. S-H. Liu and colleagues report that prolonged pre- and follow-up treatment of the left kidney with low intensity pulsed ultrasound (LIPUS) of mice with (1) IRI-induced AKI of the left kidney that progressed to CKD (the right kidney was removed on day 13, i.e., 1 day before euthanasia), and (2) in the adenosine-induced CKD model in mice (euthanasia on day 28). LIPUS therapy x 20 min. was applied daily targeting the left kidney for 5 days prior to induction of AKI/CKD (pre-treatment) in both mouse models and daily for another 14 and 28 days until euthanasia, respectively.
In the AKI/CKD model, LIPUS therapy significantly improved renal function (Cr, BUN, cystatin C), the albumin/Globulin ratio, the renal histopathology score and fibrosis, reduced elevated serum FGF23 and indoxyl sulfate levels, reversed EMT, and improved the protein levels of several apoptotic, oxidative stress and senescence related factors.
In the adenosine induced CKD model, LIPUS treatment of the left kidney (apparently only the left kidney was treated with LIPUS), renal function improved significantly Cr, BUN), and the progressive loss of body weight was halted. Renal histopathology and interstitial fibrosis scores improved (which kidney was examined?), and EMT was reversed.
These observations complement earlier studies that are cited and that demonstrated that LIPUS
Therapy is reno-protective in rodent AKI. Overall, the presented observations support the authors’ conclusion that the non-invasive LIPUS therapy has the potential of clinical utility. However, the clinical testing of LIPUS in patients with CKD will require additional, well designed pre-clinical studies. The authors need to address the following major concerns:
Major concerns:
1. IRI AKI/CKD model: in general practice, the contralateral normal right kidney is removed within 7 days of causing unilateral IRI injury to the left kidney. In the present study the normal right kidney was removed 1 day before euthanasia which cases a non-steady state at the time of euthanasia. What was the total kidney function between days 0 and 13 and 14? These data need to be presented, and an explanation of the utilized, unusual protocol must be provided. What was the histopathology of the removed right kidney?
2. IRI AKI/CKD model: no tissue data on renal angiogenesis, suppressed apoptosis and potentially deceased pro-inflammatory cytokine levels are provided. This is a major weakness of the study that must be addressed.
3. Did the animals receive daily anesthesia for the LIPUS therapy? This is a major stress factor and details of the used protocol need to be provided.
4. The urinary indoxyl sulfate to urinary Cr ratio should be provided.
5. Regarding the adenosine CKD model: it is unclear from the Methods section whether LIPUS was only focused on the left kidney. If so, it needs to be clearly stated. If only the left kidney was LIPUS treated, what was the renal histology and what were the other markers of renal EMT in the right kidney? In this case, the obtained beneficial effects in the LIPUS treated left kidney may have been the result of the induced splenic cholinergic anti-inflammatory pathway, as described by Okusa et al. (reference 12). This is a critical point that needs to be clarified and discussed.
6. The Discussion requires careful editorial revision.
Minor points:
1. Fig. 2 A and B: scale bards are not visible. Pls. correct.
Author Response
Reviewer 1:
Dr. S-H. Liu and colleagues report that prolonged pre- and follow-up treatment of the left kidney with low intensity pulsed ultrasound (LIPUS) of mice with (1) IRI-induced AKI of the left kidney that progressed to CKD (the right kidney was removed on day 13, i.e., 1 day before euthanasia), and (2) in the adenosine-induced CKD model in mice (euthanasia on day 28). LIPUS therapy x 20 min. was applied daily targeting the left kidney for 5 days prior to induction of AKI/CKD (pre-treatment) in both mouse models and daily for another 14 and 28 days until euthanasia, respectively. In the AKI/CKD model, LIPUS therapy significantly improved renal function (Cr, BUN, cystatin C), the albumin/Globulin ratio, the renal histopathology score and fibrosis, reduced elevated serum FGF23 and indoxyl sulfate levels, reversed EMT, and improved the protein levels of several apoptotic, oxidative stress and senescence related factors. In the adenosine induced CKD model, LIPUS treatment of the left kidney (apparently only the left kidney was treated with LIPUS), renal function improved significantly Cr, BUN), and the progressive loss of body weight was halted. Renal histopathology and interstitial fibrosis scores improved (which kidney was examined?), and EMT was reversed. These observations complement earlier studies that are cited and that demonstrated that LIPUS therapy is reno-protective in rodent AKI. Overall, the presented observations support the authors’ conclusion that the non-invasive LIPUS therapy has the potential of clinical utility. However, the clinical testing of LIPUS in patients with CKD will require additional, well designed pre-clinical studies.
The authors need to address the following major concerns:
Major concerns:
- IRI AKI/CKD model: in general practice, the contralateral normal right kidney is removed within 7 days of causing unilateral IRI injury to the left kidney. In the present study the normal right kidney was removed 1 day before euthanasia which cases a non-steady state at the time of euthanasia. What was the total kidney function between days 0 and 13 and 14? These data need to be presented, and an explanation of the utilized, unusual protocol must be provided. What was the histopathology of the removed right kidney?
Response: We appreciate the reviewer's comments. We designed this IRI-CKD model according to the literatures [Harwood et al., 2022 (preprint in 2020); Shu et al., 2018], which the ischemic left kidney was performed and the right nephrectomy was performed 24 h before the mice were euthanized. The study of Shu et al. has shown that the serum creatinine levels, renal histological score, and fibrosis markers protein expression levels were significantly increased in the blood samples and left kidney tissues, which were collected at 2, 7 or 14 days after left kidney ischemia and right nephrectomy was performed one day before animal euthanasia. In our preliminary experiments, we found that this IRI mouse model with right nephrectomy performed one day before animal sacrifice could effectively decrease the mortality and induced serious renal injury and fibrosis. In the present study, the ischemic left kidney was performed and the normal right kidney was removed 1 day before euthanasia. We analyzed the physiopathological alterations and changes of signaling molecules in the left kidneys, but not in the normal right kidneys.
We have revised the descriptions in the Methods and discussed this issue in the Discussion according to the suggestion of reviewer.
- IRI AKI/CKD model: no tissue data on renal angiogenesis, suppressed apoptosis and potentially deceased pro-inflammatory cytokine levels are provided. This is a major weakness of the study that must be addressed.
Response: We appreciate the reviewer's comments. We have added the data and discussion for this issue in this revised manuscript according to the suggestion of reviewer.
In the Results: (1) Figure 3B: “Moreover, the apoptosis-related signaling molecules, such as Chop, Bax, and cleaved caspase 3, could also be induced in the kidneys of IRI-CKD mice. LIPUS significantly inhibited the induction of these apoptosis-related signaling molecules (Figure 3B).” (2) Figure 4: “We further examined the involvement of inflammatory cell infiltration and cytokines/growth factor expression in the kidneys of IRI-CKD mice with or without LIPUS treatment. The protein expression levels of Ly6G (a neutrophil marker), CD68 (a macrophage marker), and inflammatory cytokine TNF-α were significantly enhanced in the kidneys (Fig. 4A) of IRI-CKD mice. LIPUS could effectively and significantly inhibit these inflammatory markers (Fig. 4A). Moreover, the levels of mRNA expression for IL-6 and TGF-β (p<0.05), but not IL-1β (p=0.072), could also be significantly increased in the kidneys of IRI-CKD mice. LIPUS could also effectively and significantly reverse the increased IL-6 and TGF-β mRNA expression (Fig. 4B). (3) We do not have data on renal angiogenesis, but will discuss this issue. TGF-β signaling has been demonstrated to be involved in the renal inflammation and angiogenesis (Gewin, 2019; Guerrero and McCarty, 2007; Kinashi et al., 2018). We also found that the renal TGF-β expression in IRI-CKD mice was increased, which could be reversed by LIPUS treatment.
In the Discussion: Growth factors, such as VEGF and TGF-β, derived from mesenchymal stem cells have been demonstrated to recruit leukocytes or repair intrinsic cells through the paracrine signals or extracellular vesicles, which may be involved in AKI repair or AKI-to-CKD transition (Gao et al., 2020). The TGF-β1 mRNA level has been found to be increased starting at 12-24 h and sustaining to 14 days after IRI in rats (Basile et al., 1996). TGF-β can trigger renal fibrosis, which may be through its action on macrophage chemoattraction; however, fibrosis may possibly be induced by macrophages via cytokines production other than TGF-β1 (Gewin, 2019). Guerrero and McCarty have indicated that TGF-β possesses the angiogenic and angiostatic properties under physiology and pathology conditions during vascular development base on its expression content and tissue/organ context (Guerrero and McCarty, 2007). Kinashi et al. have mentioned that TGF-β triggers peritoneal dialysis-associated peritoneal fibrosis and neoangiogenesis via VEGF-A interaction; a TGF-β/VEGF-C pathway may also participate in renal and peritoneal fibrosis-associated lymph-angiogenesis (Kinashi et al., 2018). The TGF-β-induced renal tubular cell apoptosis has been suggested to be due to cell cycle arrest rather than direct pro-apoptotic action (García-Sánchez et al., 2010). Therefore, the TGF-β signaling may participate in renal inflammation, apoptosis, and angiogenesis. The present study also found that the renal TGF-β expression in IRI-CKD mice was increased, which could be reversed by LIPUS treatment. However, the detailed effects and mechanisms for LIPUS on the renal angiogenesis, apoptosis, and inflammation in IRI-CKD mouse model need to be clarified in the future.
- Did the animals receive daily anesthesia for the LIPUS therapy? This is a major stress factor and details of the used protocol need to be provided.
Response: We appreciate the reviewer's comments. The mice received daily anesthesia during the LIPUS therapy. We have added this information in the Methods of this revised manuscript according to the suggestion of reviewer.
- The urinary indoxyl sulfate to urinary Cr ratio should be provided.
Response: We appreciate the reviewer's comments. Since the amount of urine collected from the mice was too small to measure, we took blood samples to measure the content of indoxyl sulfate. Serum indoxyl sulfate has been demonstrated to be associated with CKD progression or CKD-associated diseases in patients and mouse models. We discussed this issue in the Discussion of this revised manuscript.
In a cohort study of CKD patients, the levels of serum indoxyl sulfate have been found to be related to the vascular alterations and mortality [Barreto et al., 2009]. The positive relationship between serum indoxyl sulfate levels and CKD progression in pediatric patients has also been reported [Holle et al., 2020]. Yang et al. have found that the increased serum indoxyl sulfate induces the platelet hyperactivity, leading to the thrombosis in CKD mice [Yang et al., 2017]. In the present study, we also observed the increase in serum indoxyl sulfate in CKD mice, which could be effectively decreased by LIPUS treatment, implying the possibly pathological role of indoxyl sulfate in CKD mice.
- Regarding the adenosine CKD model: (1) it is unclear from the Methods section whether LIPUS was only focused on the left kidney. If so, it needs to be clearly stated. (2) If only the left kidney was LIPUS treated, what was the renal histology and what were the other markers of renal EMT in the right kidney? (3) In this case, the obtained beneficial effects in the LIPUS treated left kidney may have been the result of the induced splenic cholinergic anti-inflammatory pathway, as described by Okusa et al. (reference 12). This is a critical point that needs to be clarified and discussed.
Response: We appreciate the reviewer's comments.
(1) We have added the description for LIPUS treatment that the left kidney of the IRI mice or adenine-treated mice was given LIPUS exposure by 20 min/day in the Methods (4.2. LIPUS Treatment) according to the suggestion of reviewer.
(2) In the Results: We added the data for right kidneys. We observed that both relative left and right kidney weight were increased in adenine-CKD mice, which could be significantly reversed by LIPUS treatment (Figure 5B of this revised manuscript). Observation of gross appearance showed that the surface of both left and right kidneys in adenine group was coarse and pale, while both left and right kidneys in adenine+LIPUS group appeared similar to normal control group (Figure 6A of this revised manuscript). We further found that the fibrosis marker α-SMA protein expression and the activation of epithelial-mesenchymal transition (EMT) (decreased E-cadherin and increased vimentin protein expression) were markedly enhanced in both left and right kidneys of adenine-CKD mice, which could be significantly reversed by LIPUS treatment (Figure 7B of this revised manuscript).
(3) In the Discussion: Our results in an adenine-CKD mouse model showed that LIPUS stimulation on the left abdominal kidney position could effectively improve both left and right kidney lesions. These findings suggest that there may be endogenous protective factors transmitted from the left kidney area to the right kidney. Gigliotti et al. have tested the protective effect of a therapeutic ultrasound on renal injury in a bilateral renal IRI-induced AKI mouse model. They found that only when ultrasound stimulated the left kidney, but not the right kidney, had a markedly renal protective effect through a splenic anti-inflammatory pathway (Gigliotti et al. 2013; 2015). The protective mechanisms, including splenic anti-inflammatory pathway and others, on the right kidney in adenine-CKD mice by LIPUS treatment, which only the left kidney is stimulated, need further investigation.
- The Discussion requires careful editorial revision.
Response: We appreciate the reviewer's comments. We have revised and re-edited the Discussion in this revised manuscript according to the suggestion of reviewer.
- Minor points: Fig. 2 A and B: scale bards are not visible. Pls. correct.
Response: We appreciate the reviewer's comments. We have revised the scale bars in Fig. 2A and B of this revised manuscript according to the suggestion of reviewer.
Reviewer 2 Report
Dear Authors,
Dear Editors,
The Manuscript represents a worthy study about the application of Low-Intensity Pulsed Ultrasound (LIPUS) to suppress progression of chronic kidney disease (CKD). The previous work of the group showed positive effect of LIPUS against the acute kidney injury (AKI). This new study offers new mechanistic aspects of LIPUS to inhibit the progression of CKD in the ischemia/reperfusion injury (IRI) and adenine-induced mouse models. The wok is well designed and decently described. Some revision and moderate English editing are necessary to improve the manuscripts’ quality. Please see the detailed comments below:
Title:
“Prevents” implies complete reversion of the process. In this study, rather a reduction of senescence and EMT-associated markers was observed.
Suggestion: “Therapeutic ultrasound halts progression of CKD in vivo via mediation/regulation of markers associated with renal epithelial-mesenchymal transition (EMT or fibrosis?) and senescence”
Abstract:
l.24: Is LIPUS used clinically for soft tissues? Please double-check this information and adjust accordingly.
l.35: “…LIPUS treatment showed benefits for CDK progression…” This sentence implies that LIPUS helps the progression of CDK. Please rephrase.
Introduction:
l.46: please define the acronym ESRD.
Please elaborate on your previous work (ref. 13 and 14) and how the selected parameters (based on which tests) for LIPUS were chosen.
Please elaborate why two different CKD mouse models were used? How are they similar/different? What was the reasoning behind it?
Results:
Why the results of LIPUS effects on two CKD models were not compared to each other? Are these two models not comparable?
Why albumin/globulin was not measured for the adenine-group? In the discussion it mentions, that this ratio is a critical prediction factor for the CKD outcome.
Discussion:
Please elaborate on the size of the LIPUS transducer and size of the animal. If this were translated for clinical application, how applicable the small transducer would have been to be used for human? Please use this article (10.22203/eCM.v042a20) for your guidance.
Please elaborate on why were the mice pretreated with LIPUS before the CKD induction? What is the reasoning behind it? Would the results change, if this wasn’t undertaken?
Figures:
Figure 3: Charts’ labels are not visible at all. Please increase the font and make it readable.
Minor comments:
Low-Intensity Pulsed Ultrasound is written with a hyphen, please correct throughout the manuscript.
M&M, l. 249: “Mice were had access…”
Author Response
Reviewer 2:
Dear Authors,
Dear Editors,
The Manuscript represents a worthy study about the application of Low-Intensity Pulsed Ultrasound (LIPUS) to suppress progression of chronic kidney disease (CKD). The previous work of the group showed positive effect of LIPUS against the acute kidney injury (AKI). This new study offers new mechanistic aspects of LIPUS to inhibit the progression of CKD in the ischemia/reperfusion injury (IRI) and adenine-induced mouse models. The wok is well designed and decently described. Some revision and moderate English editing are necessary to improve the manuscripts’ quality.
Please see the detailed comments below:
- Title:
“Prevents” implies complete reversion of the process. In this study, rather a reduction of senescence and EMT-associated markers was observed.
Suggestion: “Therapeutic ultrasound halts progression of CKD in vivo via mediation/regulation of markers associated with renal epithelial-mesenchymal transition (EMT or fibrosis?) and senescence”
Response: We appreciate the reviewer's comments. We have changed the title in this revised manuscript according to the suggestion of reviewer.
- Abstract:
l.24: Is LIPUS used clinically for soft tissues? Please double-check this information and adjust accordingly.
l.35: “…LIPUS treatment showed benefits for CDK progression…” This sentence implies that LIPUS helps the progression of CDK. Please rephrase.
Response: We appreciate the reviewer's comments. We have revised these sentences in this revised manuscript according to the suggestion of reviewer.
(1) the “soft” was deleted; (2) the “benefits” was changed to “reduction”.
- Introduction:
l.46: please define the acronym ESRD.
Please elaborate on your previous work (ref. 13 and 14) and how the selected parameters (based on which tests) for LIPUS were chosen.
Please elaborate why two different CKD mouse models were used? How are they similar/different? What was the reasoning behind it?
Response: We appreciate the reviewer's comments. We have revised the manuscript for these issues according to the suggestion of reviewer.
These revised descriptions were shown in the Introduction section as follows:
“The current evidence…for CKD/end stage renal disease (ESRD) patients.”
“In this study, we investigated the protective effects of LIPUS treatment on CKD in vi-vo. Two CKD mouse models induced by both IRI and adenine administration, which both caused CKD through different pathogenic mechanisms, were used to verify the protective effects of LIPUS on CKD. In clinical applications, the intensity range of therapeutic ultrasound is generally set to 30–1000 mW/cm2 [14]. Our previous study showed that LIPUS at 100 mW/cm2 intensity, but not 30 mW/cm2, effectively improved the H2O2 or hypoxia/reoxygenation-induced acute renal cell injury in vitro and the physiopathological alterations induced by AKI in vivo [13]. Our preliminary experiments have also tested the effects of LIPUS on CKD mice with various parameters such as 30-300 mW/cm2 intensity based on the previous studies [13,14], and finally selected the 3 MHz, 100 mW/cm2, 20 minutes/day intensity for formal experiments.”
- Results:
Why the results of LIPUS effects on two CKD models were not compared to each other? Are these two models not comparable?
Why albumin/globulin was not measured for the adenine-group? In the discussion it mentions, that this ratio is a critical prediction factor for the CKD outcome.
Response: We appreciate the reviewer's comments.
(1) In this study, two CKD mouse models induced by both IRI and adenine administration, which both caused CKD through different pathogenic mechanisms, were used to verify the protective effects of LIPUS on CKD. Besides in an IRI-CKD model, we just wanted to verify the effects of LIPUS on CKD using another animal mode (adenine-induced CKD mouse model).
(2) We have added the data for albumin/globulin ratio for the adenine-induced CKD mouse model. Data were shown in Figure 5D of this revised manuscript.
- Discussion:
(1) Please elaborate on the size of the LIPUS transducer and size of the animal. If this were translated for clinical application, how applicable the small transducer would have been to be used for human? Please use this article (10.22203/eCM.v042a20) for your guidance.
(2) Please elaborate on why were the mice pretreated with LIPUS before the CKD induction? What is the reasoning behind it? Would the results change, if this wasn’t undertaken?
Response: We appreciate the reviewer's comments. We have revised the manuscript for these issues according to the suggestion of reviewer.
We have discussed these issues in the Discussion as follows:
(1) We referred to the report of Puts et al. (10.22203/eCM.v042a20) for guidance. The diameter of Sonicator®-740 ultrasound probe used our study is about 1.13 cm, which applies the exposed site to an effective radiating area of 1 cm2. When ultrasound probe was applied to the kidney of male adult mouse, whose kidney size was about 1.07 cm (Kito et al., 2017), the kidney could be coupled with the transducer. According to the instrument manufacturer's instruction manual, this pencil-shaped 1 cm2 ultrasound probe can clinically deliver ultrasound to hard-to-reach body parts as well as to smaller treatment areas.
(2) We discussed this issue in the Discussion. In the present study, the mice were pretreated with LIPUS before the CKD induction. In our preliminary tests, the results showed that when LIPUS was administered after the CKD induction in mice, the therapeutic efficacy was not as good as that of those who were given LIPUS beforehand. This may be a LIPUS parameter setting problem, and we have not yet found an appropriate parameter for post-treatment with LIPUS. We hope to continue to explore this issue in the future and list this issue as a limitation of this study.
- Figures:
Figure 3: Charts’ labels are not visible at all. Please increase the font and make it readable.
Response: We appreciate the reviewer's comments. We have revised Figure 3 in this revised manuscript according to the suggestion of reviewer.
- Minor comments:
Low-Intensity Pulsed Ultrasound is written with a hyphen, please correct throughout the manuscript.
M&M, l. 249: “Mice were had access…”
Response: We appreciate the reviewer's comments. We have corrected these issues in this revised manuscript according to the suggestion of reviewer.
Round 2
Reviewer 1 Report
Dr. Liu and colleagues submitted a revised version of their paper in which LIPUS administration to the left kidney x 20 min/day of anesthetized mice 5 days prior to and for 14 days following IRI (right nephrectomy on day 13), euthanasia on day 14 was used as a CKD model. In the adenine model of CKD, the left kidney was treated with LIPUS 5 days prior to the initiation of the p.o. adenine administration and for the following 28 days when mice were euthanized.
The LIPUS treatments resulted in improved residual renal function, maintained body weights,
Reduction in the marker for senescence and EMT, improved inflammatory profiles, less kidney injury and interstitial fibrosis, improved ant-oxidant and proinflammatory conditions and downregulation of other kidney damaging systems.
The authors have tried to respond to some of the initial concerns that were raised. However,
numerous significant experimental design, methodological, data and interpretation questions remain. The utilized CKD models and the lengthy LIPUS therapy are not adequate to support considerations for clinical translation.
The authors must address the following remaining issues:
Major:
1. Since it is very likely that the LIPUS therapy targeting the left kidney in both CKD models was exclusively or also effective via the splenic cholinergic anti-inflammatory pathway, as reported by Okusa et al. (reference 12, now cited again as reference 42). A simple chemical splenectomy could have been tested to clarify this issue.
2. Although body weights were monitored chronically, renal function prior to euthanasia should have been recorded.
3. As the LIPUS effect was only documented at 14 and 28 days, respectively, no long-term data were obtained in order to determine whether the observed beneficial effects were permeant or only transient.
4. Fig. 2Ba: no glomeruli are shown, suggesting that images were obtained from the renal medulla and not the cortex.
5. No data on the status of the renal microvasculature/vasoprotection are shown.
6. The expression of key anti-apoptotic genes, e.g., Bcl-2 etc., was not investigated.
7. The clarity of the English used in the manuscript needs improvement of the original and revised versions.
Minor:
1. Ref. 12 and 42 are identical. Pls. correct.
2. Fig. 3A: E-Cadherin expression in IRI-CKD kidneys not reported. Pls. correct.
Author Response
Dear Reviewer-1:
This study is a basic research providing information for the possibility of LIPUS as a treatment strategy for CKD. In the first-round comments, we have tried our best to revise the manuscript according to the comments of reviewer. In this second-round comments, we still do our best to revise the manuscript and respond to reviewer’s comments. However, some required experiments from the second-round comments, are really beyond the reach of this study. We sincerely hope the reviewer to understand our difficulties and can be given the opportunity to respond to these comments as the study limitations showing in the Discussion section of this revised manuscript.
Our responses for these comments are shown as follows:
Major:
- Since it is very likely that the LIPUS therapy targeting the left kidney in both CKD models was exclusively or also effective via the splenic cholinergic anti-inflammatory pathway, as reported by Okusa et al. (reference 12, now cited again as reference 42). A simple chemical splenectomy could have been tested to clarify this issue.
Response: We appreciate the reviewer's comments. Okusa et al. (reference 12) found that ultrasound prevented renal ischemia-reperfusion injury by stimulating the splenic anti-inflammatory pathway in mice; they performed an acute renal injury (AKI) mouse model with 24-48-hours duration. However, in the present study, we have demonstrated that ultrasound halted the progression of CKD in two CKD mouse models with 14- or 28-days duration that are the very different models from the study of Okusa et al. We do not rule out this possibility, which the splenic anti-inflammatory pathway is involved. Nevertheless, at present, we cannot immediately conduct animal experiments because we need to re-apply for the approval of the animal Ethics Committee of our university, and due to the ongoing covid-19 epidemic in our country, leading to that the conduct of animal experiments is restricted. We sincerely hope that the reviewer can allow us to discuss and state as a study limitation shown in the end of Discussion section (as follows).
“There are several study limitations for this study: (1) It was very likely that the LIPUS therapy targeting the left kidney in both CKD models was exclusively or also effective via the splenic anti-inflammatory pathway, as reported by Okusa et al. [12,42]; … These issues are left for future investigation.”
- Although body weights were monitored chronically, renal function prior to euthanasia should have been recorded.
Response: We appreciate the reviewer's comments. When we previously established this adenine-induced CKD animal model, we had tested the renal function (serum BUN and creatinine levels) on day 21. The results showed that renal function was moderately decreased in adenine group in this stage, which could be significantly reversed by LIPUS (BUN: sham, 28.62 ± 3.92, adenine, 127.53 ± 24.45, adenine+LIPUS, 44.84 ± 5.96 mg/dL, n=5, p<0.05; creatinine: sham, 0.33 ± 0.02, adenine, 0.98 ± 0.09, adenine+LIPUS, 0.49 ± 0.06 mg/dL, n=5, p<0.05). We added these results in this revised manuscript.
- As the LIPUS effect was only documented at 14 and 28 days, respectively, no long-term data were obtained in order to determine whether the observed beneficial effects were permeant or only transient.
Response: We appreciate the reviewer's comments. Both animal models used in this study are well-established models; the design of the testing periods are also based on reports in the literatures. In order to respond to different models, there have different time-course designs, such as Gigliotti et al. found that ultrasound prevented renal ischemia-reperfusion injury (AKI mouse model) during time course of 24-48 hours (an ultrasound protocol 24 hours before renal ischemia and after 24 hours of reperfusion). In the present study, we have demonstrated that ultrasound halted the progression of CKD in two CKD mouse models with 14- or 28-days duration that are the very different models from the study of Okusa et al. We sincerely hope the reviewer to understand our difficulties and can allow us to discuss and state as a study limitation shown in the end of Discussion section (as follows).
“There are several study limitations for this study: … (2) As the LIPUS effect was only documented at 14 and 28 days in both IRI and adenine models, respectively, no long-term data were obtained in order to determine whether the observed beneficial effects were permeant or only transient; ... These issues are left for future investigation.”
- Fig. 2Ba: no glomeruli are shown, suggesting that images were obtained from the renal medulla and not the cortex.
Response: We appreciate the reviewer's comments. We have revised the Fig. 2Ba according to the suggestion of reviewer that the glomeruli and tubules were shown.
- No data on the status of the renal microvasculature/vasoprotection are shown.
Response: We appreciate the reviewer's comments. The aim of this study was to investigate the protective effects of LIPUS treatment on CKD in vivo, focusing on epithelial-mesenchymal transition and senescence. The present study demonstrated that therapeutic ultrasound halted the progression of CKD in vivo via regulation of markers associated with renal epithelial-mesenchymal transition and senescence. We do not rule out the possibility in which renal microvasculature/vasoprotection contributes to the effects of LIPUS. Nevertheless, at present, we cannot immediately conduct animal experiments because we need to re-apply for the approval of the animal Ethics Committee of our university, and due to the ongoing covid-19 epidemic in our country, leading to that the conduct of animal experiments is restricted. We sincerely hope that the reviewer can understand our difficulties and allow us to discuss and state as a study limitation shown in the end of Discussion section (as follows).
“There are several study limitations for this study: … (3) The status of the renal microvasculature/vasoprotection is also important for LIPUS treatment... These issues are left for future investigation.”
- The expression of key anti-apoptotic genes, e.g., Bcl-2etc., was not investigated.
Response: We appreciate the reviewer's comments. The Bcl-2 protein expression was tested and shown in Fig. 3B. of this revised manuscript according to the suggestion of reviewer.
- The clarity of the English used in the manuscript needs improvement of the original and revised versions.
Response: We appreciate the reviewer's comments. This revised manuscript has been received an English editing service by MDPI (English Editing Invoice ID: english-51596).
Minor:
- Ref. 12 and 42 are identical. Pls. correct.
- Fig. 3A: E-Cadherin expression in IRI-CKD kidneys not reported. Pls. correct.
Response: We appreciate the reviewer's comments. We have corrected these issues in this revised manuscript according to the suggestion of reviewer.
(1) The Ref. 42 has been deleted.
(2) The E-cadherin protein expression has been added in Fig. 3A.
Round 3
Reviewer 1 Report
The authors have responded to the major issues that were raised.
Some basic concerns about the experimental design remain, as previously described.
English language editing is still needed.
Author Response
Dear Reviewer 1:
Thank you very much for giving us an opportunity to revise our manuscript.
In the first-round comments, we have tried our best to revise the manuscript according to the comments of reviewer. In the second-round comments, we still did our best to revise the manuscript and respond to reviewer’s comments, although some required experiments are really beyond the reach of this study. We sincerely hope the reviewer to understand our difficulties and can be given the opportunity to respond to these comments as the study limitations showing in the Discussion section of this revised manuscript. In this third-round comment, we also do our best to revise the comments by reviewer.
Our responses for these comments are shown as follows:
- The authors have responded to the major issues that were raised.
Response: We appreciate the reviewer's positive comments.
- Some basic concerns about the experimental design remain, as previously described.
Response: We appreciate the reviewer's comment. We have discussed and stated these basic concerns about the experimental design as the study limitations in the Discussion section. We will do our best to resolve these issues in the future works.
- English language editing is still needed.
Response: We appreciate the reviewer's comment. This revised manuscript has been received an English editing service by MDPI (English Editing Invoice ID: english-51596). We have re-checked and revised the manuscript throughout through an English editing service again according to the suggestion of reviewer.